# Let It Beat: How Lifestyle and Psychosocial Factors Affect the Risk of Sudden Cardiac Death—A 10-Year Follow-Up Study

**DOI:** 10.3390/ijerph19052627

**Published:** 2022-02-24

**Authors:** Jana Obrova, Eliska Sovova, Katerina Ivanova, Jana Furstova, Milos Taborsky

**Affiliations:** 1Department of Internal Medicine I—Cardiology, University Hospital Olomouc, 779 00 Olomouc, Czech Republic; milos.taborsky@fnol.cz; 2Department of Exercise Medicine and Cardiovascular Rehabilitation, University Hospital Olomouc and Faculty of Medicine and Dentistry, Palacky University Olomouc, 779 00 Olomouc, Czech Republic; eliska.sovova@fnol.cz; 3Department of Public Health, Faculty of Medicine and Dentistry, Palacký University Olomouc, 779 00 Olomouc, Czech Republic; katerina.ivanova@upol.cz; 4Olomouc University Social Health Institute, Palacký University Olomouc, 779 00 Olomouc, Czech Republic; jana.furstova@oushi.upol.cz

**Keywords:** sudden cardiac death, prevention, implantable cardioverter defibrillator, psychosocial factors

## Abstract

(1) Background: The aim of this study was to evaluate the lifestyle and occurrence of psychosocial factors in patients with a high risk of sudden cardiac death (SCD) and to explore their effect on the occurrence of the adequate therapy of an Implantable Cardioverter Defibrillator (ICD). (2) Methods: In this retro-prospective single-centre study, a group of patients aged 18–65 years old, who underwent the first ICD implantation for primary (PP) or secondary (SP) prevention between 2010–2014, was studied. The control group consisted of pair-matched (age ± 5 years, gender) respondents without a high risk of SCD. Information was obtained using a self-reported questionnaire and hospital electronic health records. The adequacy of ICD therapy was evaluated regularly until 31 January 2020. Multivariate logistic regression models were employed to assess the risk of SCD. (3) Results: A family history of SCD, coronary artery disease, diabetes mellitus and depression significantly aggravated the odds of being at a high risk of SCD. The occurrence of an appropriate ICD therapy was significantly associated with being in the SP group, BMI, education level and TV/PC screen time. (4) Conclusions: Lifestyle and psychosocial factors have been confirmed to affect the risk of SCD. Early identification and treatment of coronary artery disease and its risk factors remain the cornerstones of preventive effort. Further research is needed to evaluate the complex nature of psychosocial determinants of cardiac health.

## 1. Introduction

Sudden cardiac death (SCD) represents a serious problem due to its incidence and impact on society, especially the victims’ families [1,2]. It can be prevented by implantation of an Implantable Cardioverter–Defibrillator (ICD) [3,4,5,6,7,8]. This is a small battery-powered device placed in the chest to detect and stop any life-threatening arrhythmias in patients at high risk of SCD. Persons who have experienced symptomatic life-threatening arrhythmia or have been successfully resuscitated from sudden cardiac arrest belong to this group, and they usually undergo implantation of an ICD for “secondary prevention” (SP) [2]. Unfortunately, most people die during their first episode of malignant arrhythmia [9], which emphasises the importance of “primary prevention” (PP). However, the ability to predict future cardiac arrest remains insufficient. Current risk stratification based on left ventricle ejection fraction (LVEF) and assessment of clinical severity of heart failure has low sensitivity and specificity [10]. On the other hand, up to 21% of patients experience an inadequate shock during one to five years of follow-up, with all its psychosocial consequences [11,12,13,14]. Thus, there is a need for better risk stratification and further management of high risk SCD patients.

Psychosocial factors have been recognised as important and potentially modifiable risk factors of cardiovascular diseases (CVD). They can influence the incidence and course of multiple CVD conditions independently of other conventional risk factors [15,16,17,18,19,20,21]. The available evidence is so robust that national and international guidelines and position papers have increasingly taken such factors into account [22,23,24,25,26,27,28,29]. Studies have focused mostly on acute or chronic stress, anxiety, depression, locus of control, low socioeconomic status, low social support and isolation [12,16,17,21,30,31,32,33,34,35,36]. These factors are highly prevalent in cardiac patients [20,33]. They affect the cardiovascular (CV) system through various biological (immune, neuroendocrine) and behavioural pathways, which interact with one another in a complex manner [20,37,38]. They are associated with more adverse lifestyle behaviours (smoking, low physical activity, diet, alcohol consumption), which facilitate the development of traditional CV risk factors (obesity, hypertension, dyslipidaemia, diabetes mellitus). They may also act as barriers to lifestyle changes and treatment adherence [24,39,40,41]. Growing evidence supports a causal relationship between these factors and coronary artery disease (CAD), chronic heart failure, arterial hypertension and some arrhythmias [24]. Therefore, there is an appealing assumption that they also play a significant role in sudden cardiac death (SCD). According to prospective cohort studies, psychological factors, such as depressiveness, anxiety and the Type D personality, significantly increase the vulnerability to ventricular arrhythmias in cardiac patients. Acute psychosocial stressors (for example, a disaster or a stressful soccer match) may trigger sudden cardiac events [42,43,44].

In this study, we focused on psychosocial factors in high risk SCD patients. The aims of this study were (1) to compare the secondary and primary prevention group of patients with an ICD with the control group (without ICD) and to evaluate whether the lifestyle and psychosocial factors could affect the probability of being at high risk of SCD; and (2) to compare the primary and secondary prevention patients to assess the effect of lifestyle and psychosocial factors on the occurrence of the adequate ICD therapy during the follow-up.

## 2. Materials and Methods

### 2.1. Participants and Procedure

This is a retro-prospective single centre study of high risk SCD patients, aged 18–65 years, who underwent their first ICD implantation (primo-implantation) with or without cardiac resynchronization therapy (CRT-D) between 1 January 2010 and 31 December 2014 at the Department of Internal Medicine I—Cardiology, Olomouc University Hospital, Czech Republic. As a first step, they filled a self-reported questionnaire which retrospectively assessed the situation before the implantation and results were compared with control group. A study group was then prospectively followed until 31 January 2020 (Figure 1).

A total of 896 patients underwent ICD/CRT-D primo-implantation, 333 of whom met the age inclusion criterion of 18–65 years and remained in our outpatient care. Only 147 of them agreed with enrolment and filled out the questionnaire (44% response rate).

The sample included high-risk SCD patients with an ICD implanted for primary prevention (PP) and patients with an ICD for secondary prevention (SP). The control group consisted of age-matched (±5 years) and sex-matched persons without a high risk of SCD and thus without an indication of primary or secondary preventive ICD implantation according to valid guidelines [45]. The control sources were community-based or hospital-based (patients from non-cardiac wards/outpatient clinics and their relatives).

Information was obtained using a self-reported questionnaire and hospital electronic health records at the time of implantation. The adequacy of ICD therapy (i.e., antitachycardia pacing or shock to terminate life-threatening ventricular tachycardia or fibrillation) and complications associated with ICD implantation were evaluated during regular outpatient check-ups of the device from the time of implantation to 31 January 2020. The date and cause of death were found out from the data of the Institute of Health Information and Statistics of the Czech Republic.

### 2.2. Questionnaire

We obtained data on personal patterns of cardiovascular disease and risk factors (diabetes mellitus, hypertension, dyslipidaemia, obesity), family history of SCD, lifestyle (tobacco use, physical activity, screen time during leisure time) and education. Respondents reported suffering from depression. Psychosocial factors were evaluated using questions similar to those used in the Interheart study [16]. Three questions assessed psychosocial stress by asking about the feeling of (1) stress in personal life, (2) stress in working life and (3) financial stress. Stress was defined as a feeling of irritability, anxiety, or as having sleep difficulties as a result of conditions at work, at home, or a lack of money. For every question, participants responded how often they felt stress using one of these options: (1) never, (2) sometimes, (3) often, (4) permanently. Any question answered by option (3) often or (4) permanently were evaluated as the presence of high stress. Thus, high stress means the frequent or permanent feeling of stress in personal life and/or work life and/or financial stress. Participants were asked if they have experienced a major stressful life event in the past year, for example, major personal injury or illness, the death or major illness of a close family member, marital separation, divorce or a major conflict, loss of job or retirement, a wedding, welcoming a new family member, etc. For more information, see Appendix A.

Information on the health status of the studied group was verified in the hospital electronic health records.

### 2.3. Statistical Methods

All the statistical analyses were performed using the R software, version 4.0.5 (R Foundation for Statistical Computing, Vienna, Austria). Frequencies, percentages, means and standard deviations (SD) were used to describe the sociodemographic characteristics, lifestyle and health variables. Since the age variable was not normally distributed, a non-parametric Mann–Whitney U test was used to compare the mean age of the studied groups. The Chi-square test was employed to assess the differences between categorical variables. Multivariate logistic regression models were then employed to model the odds of (1) being a primary preventive (PP) patient, and (2) experiencing an adequate ICD therapy during the follow-up time. As predictors, we used sociodemographic characteristics, lifestyle and health variables. The significance level was set at *p* < 0.05. In addition to the *p*-values, the Cohen’s d effect size coefficients were evaluated.

## 3. Results

The descriptive characteristics of the sample are presented in Table 1. The sample comprised patients with an ICD for primary prevention (PP, *n* = 117), patients with ICD for secondary prevention (SP, *n* = 30) and the control group without ICD (*n* = 205). The mean age of the whole sample was 54.1 years; 71.0% were men. There were no significant differences in the sociodemographic, lifestyle and health variables between the PP and SP patients. Less than a quarter of PP and SP patients had completed primary education, and approximately two thirds were high school graduates. Less than 10% graduated the college/university. The majority of them spent their leisure time inactively with two or more hours of screen time a day (51% of PP patients, 66% of SP patients) and with practically no sport (78% of PP patients, 83% of SP patients). Nearly a half of them had a history of coronary artery disease. The prevalence of cardiovascular risk factors was also high—especially regarding overweight/obesity, arterial hypertension and smoking. Stressful life was reported by approximately one quarter and major life event by 40% of the study group. A relatively high incidence of SCD in family history was reported.

The control group significantly differed from the PP group in several variables; there was a higher proportion of women (χ^2^(1) = 10.1, *p* = 0.001, Cohen’s d = 0.36), fewer obese respondents (χ^2^(3) = 9.8, *p* = 0.021, Cohen’s d = 0.36), a higher proportion of college/university graduates (χ^2^(2) = 12.0, *p* = 0.003, Cohen’s d = 0.40) who, in general, spent less time at a TV/PC screen (χ^2^(3) = 13.0, *p* = 0.005, Cohen’s d = 0.41) and more often engaged in sports (χ^2^(2) = 6.4, *p* = 0.042, Cohen’s d = 0.29). The control group also reported significantly fewer health problems than the PP group (Cohen’s d 0.28–0.99). Similar significant differences also occurred between the SP and the control group (see Table 1). However, due to the low number of respondents in the SP group, the effect size of the comparisons is small, except for screen time (χ^2^(3) = 12.1, *p* = 0.007, Cohen’s d = 0.47) and coronary artery disease (CAD) (χ^2^(1) = 37.0, *p* < 0.001, Cohen’s d = 0.86).

### 3.1. Lifestyle and Psychosocial Factors in Primary Prevention Patients

In order to assess the effect of lifestyle and psychosocial factors in PP patients who are potentially at high risk of SCD, we compared the PP patients with the control group. For the purpose of these analyses, the data were pair matched according to gender and age (±5 years) of the participants. The sample for the analyses thus comprised patients with an ICD for primary prevention (PP, *n* = 117) and pair-matched respondents from the control group without an ICD (*n* = 117). Binary logistic regression was used, with the outcome variable being 1 = PP group, 0 = control group. In the multivariate logistic regression models, all factors presented in Table 1 were assessed. Initially, all the factors were included in the model and the insignificant ones were eliminated in a stepwise procedure. In the final model, participants with an SCD in their family history (OR = 2.89, CI 1.06–8.46), participants reporting CAD (OR = 9.30, CI 4.23–22.83), diabetes mellitus (OR = 2.53, CI 1.15–5.70), and those with a history of depression (OR = 7.12, CI 1.69–48.78) were significantly more likely to be in the PP group. For more information, see Table 2.

### 3.2. The Effect of Lifestyle and Psychosocial Factors on the Appropriate ICD Therapy

The focal point of this study was to evaluate if lifestyle and psychosocial factors could predict which patients would indeed benefit from ICD implantation. In other words, we assessed the effect of lifestyle and psychosocial factors on the occurrence of an appropriate ICD therapy during the follow-up time. The sample for these analyses comprised patients with an ICD for primary prevention (PP, *n* = 117) and patients with an ICD for secondary prevention (SP, *n* = 30). Overall, there were *n* = 34 (29.1%) PP patients and *n* = 14 (46.7%) SP patients who had experienced an adequate therapy of their ICD during the follow-up time. A total of 16 patients (*n* = 9 in the PP group, *n* = 7 in SP group) died. The mean follow-up times of the patients in the PP group and the SP group were 6.19 years and 6.41 years, respectively. Binary logistic regression was implemented, with the outcome variable being 1 = an appropriate ICD therapy has occurred, 0 = an appropriate ICD therapy has not occurred. Multivariate logistic regression modelling was used, with all the predictors presented in Table 1 initially included in the model. The insignificant predictors were eliminated in a stepwise procedure. In the final model, the significant predictors were the SP group, BMI, education level and time spent at a TV/PC screen during leisure time. The odds of an appropriate ICD therapy were higher in patients in the SP group (OR = 2.72, CI 1.02–7.43) and those with college/university education (OR = 7.98, CI 1.65–47.71). On the other hand, overweight or obese patients were less likely to experience an appropriate ICD therapy compared to patients with normal weight (OR = 0.28, CI 0.09–0.84 in overweight, resp. OR = 0.32, CI 0.10–0.97 in obese). Lower odds of an appropriate ICD therapy were also found in patients who spent more time at a TV/PC screen during leisure time (ORs 0.20–0.25, CIs 0.06–0.82). A borderline predictor was a stressful lifestyle, which could increase the odds of an appropriate ICD therapy by almost 2.3-fold (OR = 2.28, *p* = 0.071). Results of the final multivariate logistic regression model after stepwise exclusion of variables are presented in Table 3 and Figure 2.

## 4. Discussion

In the current study, we investigated the associations of lifestyle and psychosocial factors with the probability of being at high risk of SCD. We also explored the effect of lifestyle and psychosocial factors on the occurrence of the appropriate ICD therapy. Patients at high risk of SCD who underwent ICD implantation either for secondary prevention (after symptomatic life-threatening arrhythmia or resuscitation from sudden cardiac arrest) or primary prevention were analysed.

Patients in primary prevention of SCD dominate in our study. This is in line with world data, where shifts from secondary to primary prevention indication have been documented over the years [46,47,48,49]. Coronary artery disease is highly prevalent in SP and PP groups. Although—in correlation with recent studies—we also noticed an increase in non-ischemic cases compared to older literature [1,48]. There was a higher proportion of men in both the PP and SP groups. SCD has a large preponderance in men relative to women, probably because of the protection that women enjoy from CAD before menopause. Men have a four-fold to seven-fold greater incidence of SCD than women before 65 years of age [1]. Our control group significantly differed from the PP and SP groups in having a higher proportion of college/university graduates (16.1% vs. 7.7% or 3.3%, respectively). As the amount of college/university graduates has been constantly growing in the Czech Republic [50], we have also considered the influence of age of the patients. However, our study and control group do not differ in mean age. It has to be noted that the control group approximately reflects the proportion of university graduates among the citizens of the Czech Republic at the time of the study. Specifically, in 2012, 19% of the Czech adult population had attained tertiary education and men predominated [50]. The influence of gender can be ruled out as our higher educated control group has a higher proportion of women. Our results are in correlation with previous studies which indicate that groups with low to medium education are at a higher risk of cardiovascular events and cardiovascular death than those with higher education. A higher prevalence of risk behaviours, which facilitate the development of traditional CV risk factors, and poor health literacy could be the possible connection between low education and worse CV outcome, and our study supports this idea [51]. Two thirds of the SP and PP groups in the present study had at least one modifiable CV risk factor.

After multivariate analysis, the following parameters were significantly associated with being at a high risk of SCD (exactly in the PP group): SCD in family history, CAD, diabetes mellitus and depression. An analysis of the secondary prevention group would be the most valuable, but this was not possible due to the low number of respondents in the SP group. On the other hand, (1) the effect size of the comparisons was small but similar to those between the PP group and the control group, and (2) we did not notice any significant differences in the monitored variables between the PP and SP patients. Finally, our results are in correlation with previously published data. CAD is the leading cause of SCD [1] and diabetes mellitus increases the risk of SCD two- to ten-fold [52]. A positive family history was confirmed as a significant risk factor for SCD, and not only in inherited primary arrhythmia syndromes and cardiomyopathy [53,54,55]. Thus, international and national guidelines recommend post-mortem expert examination of all unexplained sudden death victims to investigate whether a cardiac origin should be suspected and the screening of first-degree relatives of sudden death victims [2,56,57]. Despite the recommendation, only 40% of family members are screened [2]. Cardiac arrest/SCD registries, which could improve care about survivors and initiate critical testing for them or for family members of SCD victims, are few in number in the world [58]. Our results suggest that the Czech Republic could benefit from establishing such a register. Depression was the last factor which increased the odds of being at high risk of SCD in our study, although only previously diagnosed depression was taken into account. The prevalence would have been higher if a depression screening test had been used. According to world data, depression is highly prevalent in cardiac patients (20%) and is associated with adverse cardiac events in multiple cardiovascular conditions, such as acute coronary syndrome, chronic heart failure and sudden cardiac death [59,60]. As numerous effective interventions (stress reduction techniques, cognitive behavioural therapy, pharmacotherapy and combined psychotherapy/pharmacotherapy) exist, early identification of depression (e.g., by a self-report questionnaire) could improve patient outcomes [60].

Only some patients with an ICD experienced an adequate therapy of the device. Finding the predictor of appropriate therapy among high risk SCD patients could enhance risk stratification. In the present study, a higher probability of an appropriate ICD therapy was associated with being in the SP group, normal BMI, higher education level and less screen time.

The effectiveness of secondary ICD prophylaxis has been confirmed by many studies and thus this prophylaxis is generally accepted [4,6,7]. On the other hand, the results of studies examining the effect of BMI are inconsistent [61,62,63,64,65,66]. Despite the well-known harmful effect on CV health, some studies showed a paradoxically favourable prognosis in overweight and obese cardiac patients. This phenomenon is called the “obesity paradox”, and our study supports its existence in SCD high risk patients. In addition, the higher probability of an appropriate ICD therapy in university-educated patients and less screen time is counterintuitive. In fact, out of 10 university educated patients in the PP and SP groups, seven (70%) had experienced an adequate therapy of their ICD. In the lower education groups, an adequate therapy of their ICD occurred in 34% and 27% of the elementary and high school educated patients, respectively. University educated patients with appropriate therapy were men. Most of them (*n* = 6) belonged to the PP group. They did not differ in history of CAD and CV risk factors; however, four of them had a family history of SCD. One could argue the reason for this effect might be the more stressful life of more highly educated people. In our analysis, stress was found to be a borderline predictor (*p* = 0.071). However, we examined the stress level at the time of ICD implantation only. It is known that living with ICD is associated with ongoing physical and psychosocial distress with a reduced quality of life. Higher educated people seem to have more difficulties with adjustment and coping with the new situation, and they more often regret their decision after the ICD implantation [67]. Anyway, this result needs to be confirmed on a higher number of ICD recipients, optimally in a multicentre study.

The present study brings new insight on SCD risk stratification and further management of high risk SCD patients. The strength of this study is the unique long-term follow up of SCD high risk groups. Our study has several limitations. First, it is a single-centre study with a relatively small number of respondents. Participation rate was only 44%. However, this rate is comparable or exceeds most current epidemiologic studies. Another limitation is that the information was collected only at the time of ICD implantation. Some data could have changed during the follow-up period and influenced the patients’ outcomes. Some data were only self-reported, and such reporting is subject to bias. We are aware that our study does not evaluate all psychosocial factors and can not assess their impact in all its complexity. With growing recognition of the importance of psychosocial factors as potentially modifiable cardiovascular risk factors, there is also a growing demand for standardized measures to define and quantify them. Thus, further prospective multicentre studies are needed. Larger surveys focusing on the broad social reality and psychosocial aspects of the probands’ lives could shed light on the complex associations between psychosocial factors and cardiovascular health.

## 5. Conclusions

Lifestyle and psychosocial factors were shown to affect the risk of SCD. Being at a high risk of SCD was significantly associated with SCD in family history, CAD, diabetes mellitus and depression. Early identification and treatment of coronary artery disease and its risk factors remain the cornerstones of preventive effort. The association with depression needs to be emphasized. It was shown to be a significant factor even though depression was underrepresented in our study. As numerous effective therapies exist, early identification of depression (e.g., by a self-report questionnaire) could improve patient outcomes. Family history of SCD was shown to elevate the risk as well. Systematic screening of first-degree relatives of SCD victims could reveal high risk SCD persons.

Higher probability of an appropriate therapy was associated with being in the SP group, normal BMI, higher education level and less screen time. A stressful life, which was found to be a borderline predictor, may be the connection between these results and appropriate ICD therapy. The possible moderating and/or mediating effects of distress need to be evaluated by further research focusing on the broader social reality and psychological aspects of probands’ lives.

## Figures and Tables

**Figure 1 ijerph-19-02627-f001:**
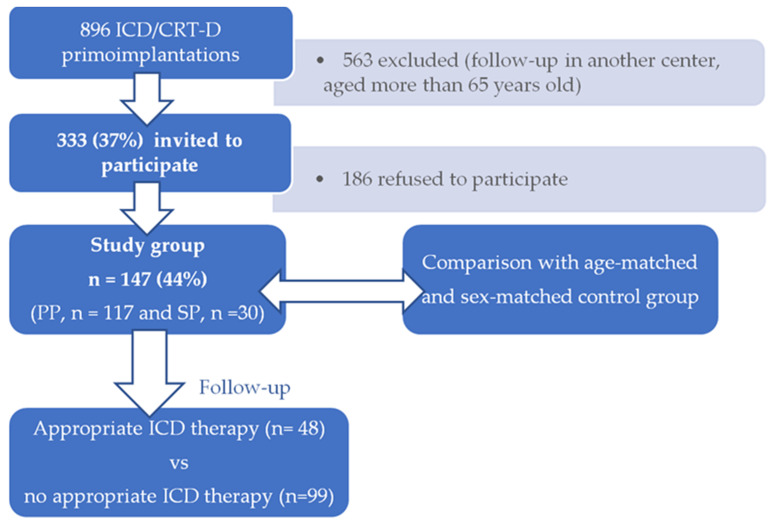
Study schema: A total of 896 patients underwent ICD/CRT-D primo-implantation between 1 January 2010 and 31 December 2014 at the Department of Internal Medicine I—Cardiology, Olomouc University Hospital, Czech Republic, 333 of whom were invited to participate. Only 147 of them agreed with enrolment, filled out the questionnaire (44% response rate) and were compared with age-matched and sex-matched control group (without high risk of SCD and ICD implantation) to evaluate whether the lifestyle and psychosocial factors could affect the probability of being at high risk of SCD. Then, they were followed up until 31 January 2020 when the effect of lifestyle and psychosocial factors on the occurrence of the adequate ICD therapy was assessed.

**Figure 2 ijerph-19-02627-f002:**
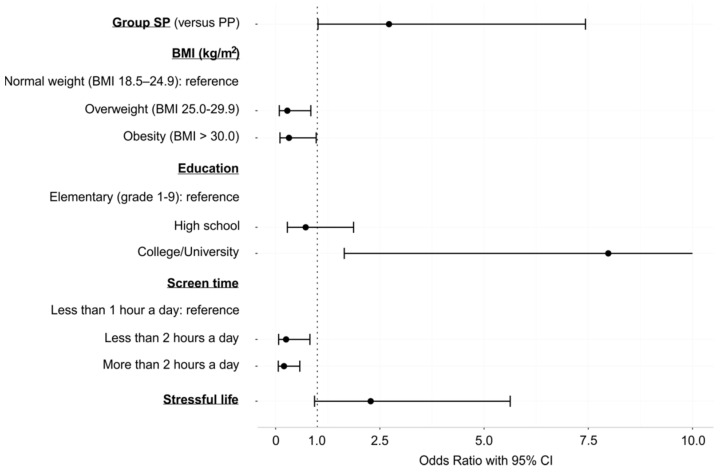
Graphical representation of the results of the final multivariate logistic regression model after stepwise exclusion of variables (Odds Ratios of the appropriate ICD therapy, and the corresponding 95% confidence interval). Note: The confidence interval of the OR for the college/university education level has been truncated at 10.0. The upper value of the interval should be 47.71, but displaying this value in the graph precludes visibility of other results in the graph.

**Table 1 ijerph-19-02627-t001:** Descriptive characteristics of the sample.

Characteristics	Primary Prevention Patients (PP)	Secondary Prevention Patients (SP)	PP vs. SP	Control Group (CG)	CG vs. PP	CG vs. SP
	*n* = 117	*n* = 30	*p*-Value	*n* = 205	*p*-Value	*p*-Value
Age: Mean (SD)			n.s.		n.s.	n.s.
	55.9 (9.7)	55.2 (8.8)		53.0 (13.4)		
Gender: *n* (%)			n.s.		0.001	n.s.
Male	95 (81.2)	23 (76.7)		132 (64.4)		
Female	22 (18.8)	7 (23.3)		73 (35.6)		
BMI: *n* (%)			n.s.		0.021	n.s.
Underweight (BMI < 18.4 kg/m^2^)	1 (0.9)	1 (3.3)		1 (0.5)		
Normal weight (BMI 18.5–24.9 kg/m^2^)	16 (13.7)	6 (20.0)		51 (24.9)		
Overweight (BMI 25.0–29.9 kg/m^2^)	41 (35.0)	15 (50.0)		77 (37.6)		
Obesity (BMI > 30.0 kg/m^2^)	58 (49.6)	8 (26.7)		68 (33.2)		
Education: *n* (%)			n.s.		0.003	0.037
Elementary (grade 1–9)	28 (23.9)	7 (23.3)		23 (11.2)		
High school	75 (64.1)	21 (70.0)		142 (69.3)		
College/University	9 (7.7)	1 (3.3)		33 (16.1)		
Screen time ^a^: *n* (%)			n.s.		0.005	0.007
Not at all	0 (0.0)	0 (0.0)		10 (4.9)		
Less than 1 h a day	19 (16.2)	3 (10.0)		44 (21.5)		
Less than 2 h a day	38 (32.5)	7 (23.3)		80 (39.0)		
More than 2 h a day	60 (51.3)	20 (66.7)		70 (34.2)		
Sport ^b^: *n* (%)			n.s.		0.042	n.s.
Less than once a week	92 (78.6)	25 (83.3)		139 (67.8)		
Once or twice a week	19 (16.2)	3 (10.0)		41 (20.0)		
At least 3x per week	5 (4.3)	2 (6.7)		24 (11.7)		
Health status ^c^: *n* (%)						
Smoking	52 (44.4)	12 (40.0)	n.s.	70 (34.1)	n.s.	n.s.
Stressful life	28 (23.9)	11 (36.7)	n.s.	59 (28.8)	n.s.	n.s.
Major life event during past year	45 (38.5)	14 (46.7)	n.s.	106 (51.7)	n.s.	n.s.
SCD in family	21 (18.0)	6 (20.0)	n.s.	14 (6.8)	0.002	0.016
Coronary artery disease	53 (45.3)	14 (46.7)	n.s.	15 (7.3)	<0.001	<0.001
Diabetes mellitus	41 (35.0)	5 (16.7)	n.s.	19 (9.3)	<0.001	n.s.
Dyslipidemia	59 (50.4)	12 (40.0)	n.s.	28 (13.7)	<0.001	0.001
Hypertension	74 (63.2)	19 (63.3)	n.s.	81 (39.5)	<0.001	0.015
Depression	14 (12.0)	3 (10.0)	n.s.	4 (2.0)	<0.001	0.015
Other diseases	63 (54.9)	12 (40.0)	n.s.	81 (39.5)	0.013	n.s.

Note: *p*-values correspond to the χ^2^ and Kruskal-Wallis tests; n.s. = non-significant (*p* > 0.05); SCD = Sudden cardiac death; ^a^ During leisure time, excluding work; ^b^ Before ICD implantation; ^c^ Self-reported health condition.

**Table 2 ijerph-19-02627-t002:** Odds of being a primary preventive (PP) patient with an ICD versus being in the control group without an ICD. Results of a final multivariate logistic regression model after stepwise exclusion of insignificant variables.

Predictor	OR	95% CI	*p*-Value
SCD in family	2.89	1.06, 8.46	0.043
Coronary artery disease	9.30	4.23, 22.83	<0.001
Diabetes mellitus	2.53	1.15, 5.70	0.022
Depression	7.12	1.69, 48.78	0.016

Note: OR = Odds Ratio; CI = Confidence Interval.

**Table 3 ijerph-19-02627-t003:** Odds of the appropriate ICD therapy. Results of a final multivariate logistic regression model after stepwise exclusion of variables.

Predictor	OR	95% CI	*p*-Value
Group SP (versus PP)	2.72	1.02, 7.43	0.047
BMI			
Underweight (BMI < 18.4 kg/m^2^)	-	-	-
Normal weight (BMI 18.5–24.9 kg/m^2^)			
Overweight (BMI 25.0–29.9 kg/m^2^)	0.28	0.09, 0.84	0.026
Obesity (BMI > 30.0 kg/m^2^)	0.32	0.10, 0.97	0.046
Education			
Elementary (grade 1–9)			
High school	0.72	0.28, 1.87	0.489
College/University	7.98	1.65, 47.71	0.014
Screen time ^a^			
Not at all	-	-	-
Less than 1 h a day			
Less than 2 h a day	0.25	0.07, 0.82	0.025
More than 2 h a day	0.20	0.06, 0.58	0.004
Stressful life	2.28	0.93, 5.63	0.071

Note: OR = Odds Ratio; CI = Confidence Interval; ^a^ During leisure time, excluding work.

## Data Availability

The dataset collected and analyzed in the current study are available from the corresponding author on reasonable request.

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
