# Peer review of "Let It Beat: How Lifestyle and Psychosocial Factors Affect the Risk of Sudden Cardiac Death—A 10-Year Follow-Up Study"

_ijerph, 2022, doi:10.3390/ijerph19052627_

Round 1

Reviewer 1 Report

The presented work touches on a very interesting and relevant topic and I congratulate the authors on doing so. For me, being not an expert in public health, some improvements need to be made.

Writing and language is very good and clear, however essential details on data acquisition and analysis are missing. Also, overall data presentation should be improved. Finally, I am missing a strong and novel take-home message, which is backed by own new data.

Some specific points (referring to line numbering in the manuscript):

  • 16 and 78: what is a retro-prospective study and how does it relate to true retrospective and prospective  studies?
  • 17: what is primo-implantation
  • 25-26: The conclusion (4) seems very generic!
  • 42: change "recognize" to "predict"
  • 44: "3-21%" seems a very wide range ... can you comment on this spread?
  • 87: The Legend for Fig 1 should be expanded with details on the process of recruiting: time period, criteria, locations, percentages ... and "met" seem to be a typo
  • 99: Please add the questionaire as a supplement
  • 104: Only three questions about psychosocial stress seem very limited and might not grasp the complexity of the situation
  • 130-146: The first part of the results (demographics) is very condensed ... try to make it more accessible
  • 162: Table 2 shows the odds of being a PP vs control .. what about the odds of being a SP?
  • Table 2: What does the "Estimate" value mean? The higher the more likely? Any quantitate interpretation possible? Any different meaning when <0?
  • 165 ff: Hard to follow which predictors are actually relevant in PP vs SP for adequate shock.
  • 168: Please define "adequate response of an ICD".
  • 187: Conclusion or brief summary of main findings should be added
  • Table 3: Is there a way to visualize the data from this table?
  • 197-198: Rephrase please. Not clear what this means!
  • 202-203: This emphasis on higher education seems a bit misleading, please put in context with other significant differences!
  • 215: "An analysis of the secondary prevention group would be the most valuable, but this was not possible due to the low number of respondents in the SP group." This seems to be a major drawback ... however I don't get what is the consequence of this ... do you present valid data on the SP? If yes, what is it? If no, why don't you leave it out?
  • 270: I am missing a conclusion on the adequate response of the ICD and not only on the risk of SCD.

Author Response

Dear reviewer,

we highly appreciate your valuable comments and expert opinion on our article. We have made  revisions to clarify all issues you have pointed. We have added missing data acquisition and analysis, and tried to improve their presentation. Our responses are in yellow, the original review text is without the highlighting.

The presented work touches on a very interesting and relevant topic and I congratulate the authors on doing so. For me, being not an expert in public health, some improvements need to be made.Writing and language is very good and clear, however essential details on data acquisition and analysis are missing. Also, overall data presentation should be improved. Finally, I am missing a strong and novel take-home message, which is backed by own new data.

Some specific points (referring to line numbering in the manuscript):

  • 16 and 78: what is a retro-prospective study and how does it relate to true retrospective and prospective studies?

Thank you for your inquiry, we have added this text to the Materials and Methods section in order to make it clearer:

As a first step, they (patients who underwent their first ICD implantation between 2010-2014) filled a self-reported questionnaire which retrospectively assessed the situation before the implantation and results were compared with control group. Study group was then prospectively followed until 31 January 2020.

  • 17: what is primo-implantation

It means the first implantation in life. We have addedthe explanation to the Materials and Methods section (line 79) and corrected line 17.

  • 25-26: The conclusion (4) seems very generic!

Thank you for your suggestion. We have further elaborated the conclusions.

  • 42: change "recognize" to "predict"

We have changed the wording according to your suggestion.

  • 44: "3-21%" seems a very wide range ... can you comment on this spread?

It is a result of analysis of 15 studies with various length of follow up (1 to 5 years) (Persson, R. Adverse Events Following Implantable Cardioverter Defibrillator Implantation: A Systematic Review. J. Interv. Card. Electrophysiol.2014, 40, 191–205, doi:10.1007/s10840-014-9913-z.)

We have changed the wording of the sentence to make it less confusing:

On the other hand, up to 21% of patients experience an inadequate shock during one to five years of follow-up, with all its psychosocial consequences [11–14].

  • 87: The Legend for Fig 1 should be expanded with details on the process of recruiting: time period, criteria, locations, percentages ... and "met" seem to be a typo

We have amended the study schema and added a legend.The legend now reads:

A total of 896 patients underwent ICD/CRT-D primo-implantation between 1 January 2010 and 31 December 2014 at the Department of Internal Medicine I – Cardiology, Olomouc University Hospital, Czech Republic, 333 of whom were invited to participate. Only 147 of them agreed with enrolment, filled out the questionnaire (44% response rate) and were compared with age-matched and sex-matched control group (without high risk of SCD and ICD implantation) to evaluate whether the lifestyle and psychosocial factors could affect the probability of being at high risk of SCD. Then they were followed up until 31 January 2020 when the effect of lifestyle and psychosocial factors on the occurrence of the adequate ICD therapy was assessed.

  • 99: Please add the questionaire as a supplement

Thank you for your suggestion. We have translated and added it as a supplement.

  • 104: Only three questions about psychosocial stress seem very limited and might not grasp the complexity of the situation

Thank you for this comment. We definitely agree. It is really hard to assess stress. We reviewed the literature and found studies with various methodology. Problem is the definition, lack of standardized measures to define and quantify the type and severity of stress and impact on studied person. Finally, we were inspired by one of the largest study addressing this question - Interheart study, which also used 3 questions - stress in personal life, at work and financial stress, which are the most common source of stress. We also asked if participants had experienced a major stressful life event in the past year (major personal injury or illness, the death or major illness of a close family member, marital separation, divorce or a major conflict, loss of job or retirement, a wedding, welcoming a new family member, etc.)

 However, we are aware of that we did not catch all aspects. We added this as a limitation of our study.

  • 130-146: The first part of the results (demographics) is very condensed ... try to make it more accessible

We have elaborated on this part of the results.

  • 162: Table 2 shows the odds of being a PP vs control ..what about the odds of being a SP?

Thank you for this question. As stated in the Introduction of the manuscript, there is a need for better risk stratification and further management of high risk SCD patients. It is clear that those patients who have already experienced a symptomatic life-threatening arrhythmia or have been successfully resuscitated from sudden cardiac arrest (secondary prevention patients) are in need of the ICD. However, the primary preventive patients who might or might not benefit from the ICD implantation need to be better stratified. In assessing the cost-benefit of the implantation, we need to know the probability those patients will benefit from the ICD implantation. That is the reason why we compare the primary preventive patients to the control group.Secondary ICD prophylaxis is well defined, effective and generally accepted. Comparison of secondary prevention group with control group could help to find other predictors (or risk score) to improve the stratification. However, due to the low number of respondents in the SP group, we were not able to perform the multivariate analyses. Nevertheless, there were no significant differences found between the SP and PP groups in the variables used for further regression analyses. The comparison of PP patients with the control group thus brings valuable information about the odds of being in higher risk of SCD.

  • Table 2: What does the "Estimate" value mean? The higher the more likely? Any quantitate interpretation possible? Any different meaning when <0?

The “Estimate” is the estimate of the beta coefficient. After exponentiating the estimate, we get the odds ratio (OR). The negative value of the estimate makes the OR below 1.

However, we have removed the estimate and the standard error values from Tables 2 and 3, as requested by Reviewer no.3.

  • 165 ff: Hard to follow which predictors are actually relevant in PP vs SP for adequate shock.

The paragraph describes the results of a multivariate regression model. Thus, all the results are interpretable for both the groups simultaneously.Like in any multivariate regression model, each coefficient is interpreted with all other predictors held constant.

  • 168: Please define "adequate response of an ICD".

We change it for more used “appropriate ICD therapy” or “adequate ICD therapy” and it means antitachycardia pacing or shock to terminate ventricular tachycardia/fibrillation (added to theParticipant and Procedures section).

  • 187: Conclusion or brief summary of main findings should be added

We have reorganized and broadened the conclusion with a summary of main findings.

  • Table 3: Is there a way to visualize the data from this table?

We have visualized the results presented in Table3 and added a new Figure 1 to the Results section.

  • 197-198: Rephrase please. Not clear what this means!

We have rephrased it (at the beginning and end of the paragraph).

  • 202-203: This emphasis on higher education seems a bit misleading, please put in context with other significant differences!

Thank you for this comment. We have put it in context with age, gender and higher prevalence of risk behaviours, which facilitate the development of traditional CV risk factors.

  • 215: "An analysis of the secondary prevention group would be the most valuable, but this was not possible due to the low number of respondents in the SP group." This seems to be a major drawback ... however I don't get what is the consequence of this... do you present valid data on the SP? If yes, what is it? If no, why don't you leave it out?

Secondary prevention is intended for persons who have already overcome circulatory arrest or life threatening arrhythmia. Secondary ICD prophylaxis is well defined, effective and generally accepted. The situation in primary prevention is more complicated. Our ability to predict future cardiac arrest is insufficient. Stratification is based on left ventricle ejection fraction (it is the volumetric fraction of fluid ejected from a chamber with each contraction, mostly measure by echocardiography) and functional capacity. The assessment of functional capacity is subjective and can change significantly over time. Severe left ventricular dysfunction is thus the strongest independent predictor of SCD. We need to find other predictors (or risk score) to improve stratification and comparison of secondary prevention group with control group could help to find it. Due to the low number of respondents in the SP group, we were not able to do multivariate analysis. Anyway, valuable is also the information that SP and PP group do not differ in our parameters. Data from SP group were also used to assess the effect of lifestyles and psychosocial factors on appropriate ICD therapy. These are our reasons not to leave them out.

  • 270: I am missing a conclusion on the adequate response of the ICD and not only on the risk of SCD.

We have added it into the conclusion.

Reviewer 2 Report

-Please specify at BMI- kg/m2

-Why dyslipidemia was not considered an important cardiovascular risk factor? This risk factor has not been analyzed.

Author Response

Dear reviewer,

we highly appreciate your valuable comments and expert opinion on our article. We have added missing data acquisition. Our responses are in yellow, the original review text is without the highlighting.

-Please specify at BMI- kg/m2

Thank you for your suggestion. We have added the units to Table 1 and 3.

-Why dyslipidemia was not considered an important cardiovascular risk factor? This risk factor has not been analyzed.

Thank you for noticing our omission of dyslipidemia. Dyslipidemia is definitely an important cardiovascular risk factor. We have analyzed it (see "2.2 Questionnaire") and forgot to include it in results. We have added it to Table 1 only as it was not a significant predictor in the multivariate regression analyses.

Reviewer 3 Report

This is an interesting study about the impact of lifestyle and psychosocial factors on SCD. The study is simple and well performed. However it is of paramount importance better describe the control population. So please specify what does it means for control population “without a high risk of SCD” which were inclusion criteria for the control group?

In table 1 how was the variable “stressful life” expressed considering that in methods it seems to be a complex domain with many variables?

I would remouve columns estimate and standard error by table 3, maintain only OR with confidence interval and p values.

Author Response

Dear reviewer,

we highly appreciate your valuable comments and expert opinion on our article. We have made  revisions to clarify all issues you have pointed. Our responses are in yellow, the original review text is without the highlighting.

This is an interesting study about the impact of lifestyle and psychosocial factors on SCD. The study is simple and well performed. However it is of paramount importance better describe the control population. So please specify what does it means for control population “without a high risk of SCD” which were inclusion criteria for the control group?

We have adapted the sentence in the article: “The control group consisted of age-matched (± 5 years) and sex-matched persons without a high risk of SCD and thus without any indication of primary or secondary preventive ICD implantation according to valid guidelines [45].”

Primary and secondary preventive ICD implantation is explained in the Introduction.

In table 1 how was the variable “stressful life” expressed considering that in methods it seems to be a complex domain with many variables?

It is really hard to assess stress. We were inspired by the multicenter Interheart study, which also used 3 questions - stress in personal life, at work and financial stress.The assessment processhas been explained in the Methods section: “For every question, participants responded how often they felt stress using one of these options (never/sometimes/often/permanently). In our study, any question answered by option “often” or “permanently” were evaluated as the presence of high stress. Thus, high stress means the frequent or permanent feeling of stress in personal life and/or work life and/or financial stress.”

I would remove columns estimate and standard error by table 3, maintain only OR with confidence interval and p values.

The tables 2 and 3 were rephrased according to the suggestion.
